Additional sauropod dinosaur material from the Callovian Oxford Clay Formation, Peterborough, UK: evidence for higher sauropod diversity

Holwerda Femke M. f.holwerda@lrz.uni-muenchen.de 1 2
Evans Mark 3 4
Liston Jeff J. 1 5 6
1 Staatliche Naturwissenschaftliche Sammlungen Bayerns (SNSB), Bayerische Staatssamlung für Paläontologie und Geologie , Munich , Germany
2 Faculty of Geosciences, Utrecht University , Utrecht , Netherlands
3 New Walk Museum and Art Gallery, Leicester Arts and Museums Service , Leicester , United Kingdom
4 University of Leicester Centre for Palaeobiology Research, School of Geography, Geology and the Environment, University of Leicester , Leicester , United Kingdom
5 Department of Natural Sciences, National Museums Scotland , Edinburgh , Scotland
6 Vivacity-Peterborough Museum , Peterborough , United Kingdom
Wedel Mathew
Electronic publication date: 2019 Feb 14
Publication date: 2019
Volume: 7
Electronic Location ID: e6404
Received 2018 Mar 17; Accepted 2019 Jan 7
Copyright: ©2019 Holwerda et al.
Copyright year: 2019
Copyright holder: Holwerda et al.
License: This is an open access article distributed under the terms of the Creative Commons Attribution License, which permits unrestricted use, distribution, reproduction and adaptation in any medium and for any purpose provided that it is properly attributed. For attribution, the original author(s), title, publication source (PeerJ) and either DOI or URL of the article must be cited.
License URL: https://creativecommons.org/licenses/by/4.0/

Keywords: Eusauropoda, Neosauropoda, Oxford Clay Formation, Middle Jurassic, Callovian, Dorsal, Caudal

Funding: The authors received no funding for this work.

==============================
Four isolated sauropod axial elements from the Oxford Clay Formation (Callovian, Middle Jurassic) of Peterborough, UK, are described. Two associated posterior dorsal vertebrae show a dorsoventrally elongated centrum and short neural arch, and nutrient or pneumatic foramina, most likely belonging to a non-neosauropod eusauropod, but showing ambiguous non-neosauropod eusauropod and neosauropod affinities. An isolated anterior caudal vertebra displays a ventral keel, a ‘shoulder’ indicating a wing-like transverse process, along with a possible prespinal lamina. This, together with an overall high complexity of the anterior caudal transverse process (ACTP) complex, indicates that this caudal could have belonged to a neosauropod. A second isolated middle-posterior caudal vertebra also shows some diagnostic features, despite the neural spine and neural arch not being preserved and the neurocentral sutures being unfused. The positioning of the neurocentral sutures on the anterior one third of the centrum indicates a middle caudal position, and the presence of faint ventrolateral crests, as well as a rhomboid anterior articulation surface, suggest neosauropod affinities. The presence of possible nutrient foramina are only tentative evidence of a neosauropod origin, as they are also found in Late Jurassic non-neosauropod eusauropods. As the caudals from the two other known sauropods from the Peterborough Oxford Clay, Cetiosauriscus stewarti and an indeterminate non-neosauropod eusauropod, do not show the features seen on either of the new elements described, both isolated caudals indicate a higher sauropod species diversity in the faunal assemblage than previously recognised. An exploratory phylogenetic analysis using characters from all four isolated elements supports a basal neosauropod placement for the anterior caudal, and a diplodocid origin for the middle caudal. The dorsal vertebrae are an unstable OTU, and therefore remain part of an indeterminate eusauropod of uncertain affinities. Together with Cetiosauriscus, and other material assigned to different sauropod groups, this study indicates the presence of a higher sauropod biodiversity in the Oxford Clay Formation than previously recognised. This study shows that it is still beneficial to examine isolated elements, as these may be indicators for higher species richness in deposits that are otherwise poor in terrestrial fauna.

Introduction

Sauropods are represented in the Middle Jurassic of the UK by two named species thus far: the Bajocian—Bathonian Cetiosaurus oxoniensis (Phillips, 1871; Owen, 1875) and the Callovian Cetiosauriscus stewarti (Charig, 1980; Charig, 1993). Cetiosauriscus is known from material found in the Peterborough Oxford Clay, and has thus far not been encountered from other localities (Woodward, 1905; Heathcote & Upchurch, 2003; Noè, Liston & Chapman, 2010). The type material comprises of a posterior dorsal vertebra, a partial sacrum, a partial caudal axial column, forelimb and partial pectoral girdle, hindlimb, and a partial pelvic girdle (Woodward, 1905). Thus far, it is recovered in phylogenetic analyses as a non-neosauropod eusauropod (e.g.,  Heathcote & Upchurch, 2003; Rauhut et al., 2005; Tschopp, Mateus & Benson, 2015, although in the last analysis, in some trees it is recovered as a basal diplodocoid as well). Another species of Cetiosauriscus, Cetiosauriscus greppini, is known from Switzerland; however, this specimen is from the Late Jurassic, and moreover, has recently been reidentified as a putative basal titanosauriform (Schwarz, Wings & Meyer, 2007).

In addition to Cetiosauriscus, four anterior caudal vertebrae (NHMUK R1984) are known from the Oxford Clay Formation. These were previously ascribed to a brachiosaurid (Upchurch & Martin, 2003; Noè, Liston & Chapman, 2010), and have more recently been reidentified as an indeterminate non-neosauropod eusauropod (Mannion et al., 2013). Another sauropod fragment from the Oxford Clay Formation is a partial distal tail segment including ten posterior(most) caudals, which was initially assigned to a diplodocid (Upchurch, 1995; Noè, Liston & Chapman, 2010). However, more recently Whitlock (2011) showed the moderate elongation of these elements to not be conclusive of placement within Diplodocoidea, and furthermore, Mannion et al. (2012) suggested a tentative placement of Neosauropoda indet., later more cautiously proposed as eusauropod indet (P Mannion, pers. comm., 2018). A partial pelvic girdle, dorsal rib and dorsal centrum NHMUK R1985-1988 (Noè, Liston & Chapman, 2010), referred to ‘Ornithopsis leedsi’ (Hulke, 1887; Woodward, 1905) from the lower Callovian Kellaways Formation, were recently referred to an indeterminate non-neosauropod eusauropod (Mannion et al., 2013). Finally, three undiagnosed ‘camarasaurid’ sauropod teeth (Martill, 1988), tentatively ascribed to Turiasauria ( Royo-Torres & Upchurch, 2012) are known from the Oxford Clay. See Table 1 for a list of sauropod material from the Oxford Clay Formation.

Table 1 Oxford Clay Formation sauropod material.

Collection reference	Material	Diagnosis	
NHMUK R1967	10 posterior caudal vertebrae	Non-neosauropod eusauropod indet	
NHMUK R1984	4 anterior caudal vertebrae	Non-neosauropod eusauropod indet	
NHMUK R1985	Left and right pubis	Non-neosauropod eusauropod indet	
NHMUK R1986	Dorsal centrum (w/o neural arch)	Non-neosauropod eusauropod indet	
NHMUK R1987	Dorsal rib	Non-neosauropod eusauropod indet	
NHMUK R1988	Left and right ischium	Non-neosauropod eusauropod indet	
NHMUK R3078	posterior dorsal vertebra, a partial sacrum, a partial caudal axial column, forelimb and partial pectoral girdle, hindlimb, and a partial pelvic girdle	Cetiosauriscus stewarti	
NHMUK R3377	3 isolated teeth	?Turiasauria	

The Middle Jurassic (Callovian) Oxford Clay Formation, UK, has yielded many marine vertebrates (ichthyosaurs, pliosaurids, cryptoclidids and other plesiosaurians, marine crocodylomorphs, sharks, and fishes (Andrews, 1910; Andrews, 1913)), as well as invertebrates (Leeds, 1956). Land-dwelling vertebrates such as dinosaurs, however, are rare from this marine setting. The Jurassic Gallery of the Vivacity-Peterborough Museum in Peterborough, and the New Walk Museum and Art Gallery in Leicester house some of these dinosaur specimens from the Oxford Clay of Peterborough. The material consists of isolated partial elements of a stegosaur, and several isolated sauropod fossils, including a two associated dorsal vertebrae, a partial anterior caudal vertebra and a partial middle caudal vertebra. These elements have been submerged in seawater; however, they do display some characters which may be used for diagnosis.

Despite the locality being a classic site for fossils, and many historical finds of marine reptiles having been described and redescribed, the sauropod fauna from the Oxford Clay has not received much attention thus far. Though associated material such as Cetiosauriscus is scarce, isolated material can be studied in detail and reveal information on both morphology and species diversity, which is important for material from the Middle Jurassic of the United Kingdom, as this is relatively scarce (Manning, Egerton & Romano, 2015). Therefore, we here describe two isolated sauropod dorsal vertebrae, as well as two isolated caudal vertebrae from the collections of the Vivacity-Peterborough Museum and of the New Walk Museum of Leicester, all from the Oxford Clay Formation of Peterborough, United Kingdom (and previously indexed in collections under ‘Cetiosaurus’), and compare them to contemporaneous and other sauropod remains.

Materials & Methods

Systematic Paleontology

Dinosauria (Owen, 1842)	
Saurischia (Seeley, 1888)	
Sauropoda (Marsh, 1878)	
Eusauropoda (Upchurch, 1995)	
?Neosauropoda (Bonaparte, 1986a)	

Geological and historical setting

The two dorsal vertebrae PETMG R85 were found in 1922 by Mr. P.J. Phillips, at London Road, Peterborough, most likely indicating the vertebrae were from the vicinity of either the Woodston or Fletton pits, to the west and east of that roadway (see Fig. 1). The ammonite embedded on the specimen is likely a Kosmoceras jasoni, a common ammonite of the Oxford Clay Formation (J Cope, pers. comm., 2018; Hudson & Martill, 1994).

Figure 1 Geographical position of King’s Dyke, Orton and Star Pit, Whittlesey, UK.

(adapted after Hudson & Martill (1994), with notes from Liston (2006)).

Details on the provenance of the caudal specimen PETMG R272 are sparse, save that it is recorded as being from the King‘s Dyke pit (see Fig. 1). No date of discovery is known. However, the King’s Dyke pit first opened in 1969 (Hillier, 1981). Stratigraphically, this pit ranges from the lower Athleta, Phaeinum Subchronozone, down to the Kellaways Sand (Lower Callovian Calloviense Chronozone, K Paige, pers. comm., 2018), which is further supported by identifications of bivalves on PETMG R272 as Eonomia timida (T Palmer, pers. comm., 2018). Although LEICT G. 418.1956.21.0 is recorded as being from the Peterborough Oxford Clay Formation, its precise provenance is unknown. The original label on the specimen dates from 1956, when a number of brick pits were active, including parts of the Orton, Fletton, Farcet and Yaxley pits (Hillier, 1981, see Fig. 1). In addition, there would also be the worked out pits that would be accessible for collectors to search the pit faces and spoil heaps thereof. The strata of all the Peterborough clay pits extend from the Kellaways Formation up to the Stewartby Member of the Peterborough Formation (see Hudson & Martill, 1994, for a more detailed geological setting), and therefore date exclusively to the Callovian (Middle Jurassic, ∼155 Ma).

Results

Morphology

Dorsal vertebrae PETMG R85

The two associated dorsal vertebrae PETMG R85 (Figs. 2 and 3) are incomplete; the first dorsal has the centrum and a small part of the neural arch preserved; the second dorsal only the centrum. Both dorsal elements are partially covered in sediment, probably clay, and are covered with marine invertebrates, showing long-time immersion in seawater. The position of the dorsals is unclear; however, the relative dorsoventral length compared to the anteroposterior length of the centra suggests a more posterior position.

Figure 2 Posterior dorsal PETMG R85.

In anterior (A), posterior (B), ventral (C), dorsal (D), right lateral (E) and left lateral (F) views. Scalebar is 10 cm.

Figure 3 Posterior dorsal PETMG R85.

In anterior (A), posterior (B), left lateral (C), right lateral (D), ventral (E) and dorsal (F) views. Scalebar is 10 cm.

The first dorsal shows an oval anterior articular surface, which is dorsoventrally higher than transversely wide, and measures 24,7 by 21,4 cm. The anterior surface (Fig. 2A) is slightly convex, whereas the posterior surface (Fig. 2B), which is also dorsoventrally longer than transversely wide, is flat to concave, rendering the centrum very slightly opisthocoelous. The posterior articular surface measures 21,3 by 18,3 cm, and shows circular striations on the surface not covered by sediment. The anterior articular surface shows several small bivalves embedded in the matrix covering it, as well as an ammonite (Fig. 2A), see Geological Setting. It also displays a rim, ‘cupping’ the articular surface, which is also visible in lateral view (Figs. 2E and 2F). The anterior ventral surface projects further ventrally than the posterior side. In ventral view, the centrum displays rugose anteroposterior striations, as well as a slight constriction of the ventral surface, bordered by two low ridges (Fig. 2C). Furthermore, the ventral surface shows several bivalves and small pneumatic foramina. In lateral view, the centrum also shows small pneumatic or nutrient foramina (Fig. 2F). Pleurocoels are not visible, only very shallow fossae ventral to the neural arch. The centrum measures 7,6 cm long anteroposteriorly in right lateral view, and 10,8 cm in left lateral view, displaying some mild distortion, which is also visible in ventral view (Figs. 2C, 2E, 2F).

The neural arch on the first dorsal in anterior view shows the neural canal to be covered with sediment, making it unclear how large or what shape the neural canal originally was (Fig. 2A). The posterior neural canal shows the same sedimentary infill (Fig. 2B). As the infill here follows a specific shape, however, it is possible that the neural canal was oval, and dorsoventrally higher than transversely wide, both in anterior and posterior view. Lateroventral to the neural canal, rugosities extend to the base of the diapophyseal laminae; it is unclear what these rugosities are. Dorsolateral to the neural canal, possible prezygapophyseal bases are visible. Ventral to these, the base of the diapophyses is seen, which would project strongly dorsolaterally (Fig. 2A). A lip-like structure is seen dorsal to the neural canal, which is also visible in lateral (Fig. 2E) and dorsal view (Fig. 2D). Dorsal to this structure, a rugose triangular hypanthrum is seen, flanked by two ridges which might be spinoprezygapophyseal laminae (sprl, sensu Wilson, 1999). The posterior neural arch also shows the diapophyseal base to project dorsolaterally (Fig. 2B). A similar rugose triangular process is seen dorsal to the posterior neural canal, possibly the rudimentary hyposphene (Fig. 2B). Here too, this structure is flanked by two ridges, possibly the spinopostzygapophyseal laminae (spol). Lateral and ventral to this structure, two wide laminae are seen to project dorsolaterally, these could be the centropostzygapophyseal laminae (cpol), which are also visible in lateral (Figs. 2E and 2F) and dorsal (Fig. 2D) view. In lateral view, the base of the diapophyses are supported by both an anterior and posterior centrodiapophyseal lamina (acdl, pcdl). In right lateral view, a possible small centrodiapophyseal fossa (cdf) is seen (Fig. 2E). Finally, a possible spinodiapophyseal lamina (spdl) is seen to project dorsally to the base of the neural spine (which is not preserved) in both lateral views (Figs. 2E and 2F). The base of the neural spine is seen to project dorsally and slightly posteriorly, making it possible that the neural spine also projected dorsally and posteriorly. In dorsal view, the base of the spine has an oval to rhomboid shape, and is transversely wider than anteroposteriorly long (Fig. 2D).

The second dorsal centrum of PETMG R85 (Fig. 3) is preserved without any remnants of the neural arch. The centrum is amphicoelous/amphiplatyan. Neurocentral sutures are tentatively present on each lateral side of the centrum, however; these are also embedded in sediment. One is slightly visible in dorsal view (Fig. 3F). The anterior articular surface (Fig. 3A) measures 19,4 cm dorsoventrally and 19,3 cm transversely, and projects slightly further ventrally than the posterior side (Figs. 3C and 3D). It is round in shape, and shows a small ventral indentation, which could be due to taphonomic damage. The surface is covered in matrix, which embeds ammonite and belemnite remains, as well as bivalves, indicating immersion in seawater; see Geological Setting. The posterior articular surface (Fig. 3B) is more oval in shape, and dorsoventrally longer (17,7 cm) than transversely wide (13,9 cm). This surface shows rounded striations around the rim, as in the other dorsal. The ‘true’ surface is partially visible and shows a pitted central surface, whereas a part of the posterior side is also embedded in matrix and bivalves. The centrum furthermore shows no pleurocoels, only very shallowly concave areas below the possible neurocentral sutures. The surface is covered in shallow, oval nutrient or pneumatic foramina, as in the other dorsal. In ventral view, the centrum is slightly constricted transversely, and is concave, with both articular surfaces fanning out transversely from this constriction. Ventrally, also nutrient or pneumatic foramina are visible. The ventral surface of the centrum shows longitudinal striations.

Anterior caudal vertebra PETMG R272

The anterior caudal PETMG R272 (See Fig. 4) measures a maximum of 27,2 cm dorsoventrally and 26,5 cm transversely. The anterior articular surface measures 23,1 by 24,7; the posterior 25,6 by 21,8. The centrum is 15,3 cm long anteroposteriorly. It is covered in bivalves which are embedded on the surface of the bone (see Fig. 4), demonstrating long-term submersion in seawater and possible epibiont activity (Martill, 1987; Danise, Twitchett & Matts, 2014). The neural spine is missing, as well as the entire left transverse process; the right transverse process is partially preserved at its base. The centrum is transversely wider at its dorsal side than at the ventral side, and the posterior side protrudes further ventrally than the anterior side. The relative axial compression of the centrum, together with the apparent connection between the neural arch and base of the transverse processes (as far as can be seen) shows this vertebra to be one of the anterior-most caudals.

Figure 4 Anterior caudal PETMG R272.

In anterior (A), posterior (B), lateral (C), ventral (D), and dorsal (E) views. Scalebar is 10 cm.

In anterior view (Fig. 4A), the articular surface of the centrum is oval to round, and is transversely wider than dorsoventrally high. The outer surface of the articular surface is convex and displays circular striations, as is common for weightbearing bones in sauropods (F Holwerda, pers. obs., 2018). The internal ±1/3rd of the anterior articular surface is shallowly concave. The entire articular surface is ‘cupped’ by a thick rim, which mostly follows the oval to round contour of the articular surface, however, it is flattened ventrally, and on the dorsal rim it shows a slight indentation, rendering the dorsal rim heart-shaped. This rim is also seen in lateral view (Fig. 4C). In posterior view (Fig. 4B), the articular surface is heart-shaped to triangular: the ventral rim ends in a transversely pointed shape, whereas the dorsal rim shows a rounded depression on the midline, flanked by parallel convex bulges. The articular surface itself is concave, with an additional depression in the mid ±1/3rd part of the surface. The posterior articular surface is less rugosely ‘cupped’ by its rim than the anterior one.

In ventral view (Fig. 4D), the posterior rim of the centrum shows rudimentary semilunar shaped chevron facets, which are not seen on the anterior side. The transverse processes are visible as triangular protrusions that project laterally. Below each is a small oval depression. The lateral sides of the centrum are constricted, and flare out towards the anterior and posterior sides. A keel-like structure can be seen on the ventral axial midline of this vertebra. This keel is not visible as a thin protruding line, but more as a broad band protruding slightly ventrally from the ventral part of the centrum. It is possible this keel is formed by the close spacing of the ventrolateral rims of the centrum, as is described for neosauropod anterior caudal vertebrae by Harris (2006). In lateral view, the transverse processes are visible as triangular protrusions that project laterally. They are oval in cross-section. Below each is a small, oval, shallow depression. The lateral sides of the centrum are constricted, and flare out towards the anterior and posterior sides.

The anterior side of the neural canal and the base of the neural arch are set in a dorsoventrally high, anteroposteriorly flattened sheet of bone, consisting of the spinodiapophyseal/prezygodiapophyseal and centrodiapophyseal laminae, which give the neural arch (without transverse processes and neural spine) a roughly triangular shape (Fig. 4A). In particular, the high projection on the neural arch of the diapophyseal laminae suggest the existence of a ‘shoulder’, which would make the transverse processes wing-shaped (see Gallina & Otero, 2009). However; as the neural arch is incomplete, there is no certainty about the exact shape of the transverse processes and their connection to the neural arch. The neural canal is broadly arched (measuring 3,3 cm by 3,8 cm). Its dorsal rim is overshadowed by a lip-like, triangular protrusion, which could be a remnant of the hypantrum (Fig. 4A). Right above this lip-like process, a rugosely striated lamina persists along the dorsoventral midline of the neural arch, up to the dorsal-most rim of the specimen. This may possibly be the scar of a rudimentary single intraprezygapophyseal lamina (stprl, Fig. 4A). The posterior side of the neural canal is more teardrop-shaped, and is set within the neural arch, which displays shallow depressions on both sides of the neural canal; these could be small postzygapophyseal spinodiapophyseal fossae pocdf, sensu Wilson et al., 2011, Fig. 4B). Directly above it, the rami of the bases of the postzygapophyses are clearly visible. The postzygapophyses are rounded to triangular in shape (Fig. 4B). A deep oval depression is seen between them; this could be the remnant of the spinopostzygapophyseal fossae (spof, sensu Wilson et al., 2011, Fig. 4B). Finally, a V-shaped striated process is seen between the two postzygapophyses, which could be the remnant of the hyposphene.

The transverse processes appear like rounded protuberances, seen in anterior and lateral view (Figs. 4A and 4C). The ventral sides of the bases of both transverse processes are concave. In lateral view, the transverse process has a rounded to triangular shape, and is axially wider ventrally than dorsally. It is dorsally supported by a spinodiapophyseal lamina (spdl, Fig. 4E), and seems to have an anterior centrodiapophyseal lamina (acdl); however, a posterior centrodiapophyseal lamina (pcdl) is not clearly visible.

Middle caudal vertebra LEICT G.418.1956.21.0

The middle caudal LEICT G.418.1956.21.0 (Fig. 5) is an isolated element, and has no connection with the anterior caudal. Unlike the anterior caudal, this middle caudal centrum is well-preserved, with minute details clearly visible. The neural arch and neural spine are not preserved, and as the unfused neurocentral sutures show, the animal this caudal belonged to, was not fully grown (Brochu, 1996) and probably in Morphological Ontogenetic Stage 2 (MOS 2), rather than MOS 1, given the large size (sensu Carballido & Sander, 2014).

Figure 5 Middle caudal Leict LEICT G.418.1956.21.0.

In anterior (A) right lateral (B), posterior (C), left lateral (D), dorsal (E), ventral (F) views. Scalebar 10 cm.

The centrum is 21,9 cm long axially, its anterior maximum tranverse width is 21,7 cm and its posterior maximum width 18,6 cm, with posterior maximum height at 15,2 cm, giving an average Elongation Index (aEI, sensu Chure et al., 2010) of 1,31. The centrum is rectangular in shape, seen in dorsal (Fig. 5E) and ventral view (Fig. 5F), with mildly flaring anterior and posterior lateral ends of the articulation surfaces. In lateral view (Figs. 5B and 5D), the posterior ventral side protrudes further ventrally than the anterior ventral side. However, the anterior dorsal side projects further dorsally than the posterior side. Transverse processes are only rudimentarily present, as oval, rugose, lateral bulges.

The anterior articular surface is rhomboid (hexagonal to almost octagonal) in shape (Fig. 5A); the dorsal 1/3rd shows a wide transverse extension of the articular rim, whilst the lower 1/3rd shows a much narrower width, with sharply beveled constrictions between them. The ventral side shows a rounded indent on the midline, giving this articular surface a heart-shaped ventral rim. The rim itself is about 2–3 cm thick, shows concentric striations, and protrudes slightly anteriorly. The inner articular surface is flat to concave, however, the kernel shows a rugose rounded protrusion of bone, with a transverse groove running through it. The morphology of the posterior articular surface (Fig. 5C) is much more simple, oval in shape, and is wider transversely than dorsoventrally high. The articular rim is less thick than anteriorly; about 1–2 cm. The articular surface is mildly concave, with a dorsal slightly convex bulge, which is common in non-neosauropod eusauropods (e.g., Cetiosaurus, Patagosaurus (F Holwerda, pers. obs., 2011)). The dorsal side of the centrum (Fig. 3E) shows well-preserved and unfused neurocentral sutures, which span approximately the anterior 2/3rds of the axial length of the centrum. The ventral half of the neural canal is clearly visible, and shows four axially elongate, deep nutrient foramina embedded within the posterior half of the centrum. A further two shallow nutrient foramina are visible.

The ventral side of the centrum (Fig. 5F) shows two sets of chevron facets, the posterior ones of which are more pronounced. Several rugose striations run along the axial length of the ventral surface, probably for ligament attachments. Along the midline, a ventral hollow (possibly the ventral longitudinal hollow, but this is not clear) runs anteroposteriorly, braced on each lateral side by a rounded, slightly protruding beam. On each lateral side of these, shallow oval asymmetrical depressions are visible; these are caused by preparing away sediment and debris, and could possibly be fossae, but this is uncertain. Two faint ventrolateral crests are also possibly present, also visible in right lateral view (Fig. 5B). The crests are not pronounced, and on the left lateral side (Fig. 5D) the crest does not run for the entire anteroposterior length. The right lateral side (Fig. 5B) furthermore shows a faint longitudinal ridge, however, in left lateral view (Fig. 5D), this ridge does not persist on the entire lateral side of the centrum.

The lateral side of the centrum further shows several small nutrient foramina. Faint ridges are visible anterodorsal to the transverse processes, which could be vestigial diapophyseal laminae. Finally, very shallow oval depressions, possibly pneumatic, are seen ventral to the bulges of the transverse processes.

Phylogenetic framework

To explore possible phylogenetic relationships, the material studied here is used as separate Operational Taxonomic Units (OTU’s). The morphological characters of both dorsals and both caudals of this study were coded in an existing sauropod-based matrix from Carballido et al. (2017). in Mesquite (Maddison & Maddison, 2010) using non-neosauropod eusauropods as well as neosauropods. A second analysis used the diplodocoid-based datamatrix from Tschopp & Mateus (2017). See supplementary material of Tschopp, Mateus & Benson (2015), for the character matrix, explanatory notes, and references therein. See Supplementary file for this manuscript for both datamatrices including our coding. Only dorsal characters were coded for PETMG R85, anterior caudal characters could be coded for PETMG R272, and only anterior to middle, and middle to posterior characters could be coded for LEICT G.418.1956.21.0. Next to these codings, the anterior and middle caudals of Cetiosauriscus stewarti were recoded, based on the descriptions of Woodward (1905) and Charig (1980) and based on pictures of NHMUK R3078 which resulted in some character changes. See Supplemental Information 1 for our character matrix, adapted from Tschopp, Mateus & Benson (2015).

Both matrices were analysed using TNT (Goloboff, Farris & Nixon, 2008; Goloboff & Catalano, 2016) using TBR, which yielded 15,636 trees. The strict consensus tree shows the dorsals PETMG R85 as grouping with Europasaurus, and both PETMG R272 as well as LEICT G.418.1956.21.0 to be sister groups, placed within Macronaria, and sister-group to Diplodocoidea (see Fig. 6A). It should be noted, however, that PETMG R85 is unstable in this analysis, and it only takes a few more steps to move these to other nodes in the tree. Moreover, most synapomorphies for the nodes were only applicable to a few caudal characters, which may not be explicit enough for the isolated material of this study.

Figure 6 Phylogenetic analyses.

Strict consensus tree based on Carballido et al. (2017) (A) and second analysis based on Tschopp & Mateus (2017) (B) with revised Cetiosauriscus (purple) coding, and additionally PETMG R85 (orange) PETMG R272 (blue) and LEICT G.418.1956.21.0 (red) as OTU’s.

The second analysis using the matrix of Tschopp & Mateus (2017), using New Technology search recovers four trees where PETMG R272 groups with Cetiosauriscus in Diplodocimorpha, the dorsals PETMG R85 as sister-group to Diplodocidae, and finally LEICT G.418.1956.21.0 as jumping between grouping with Diplodocinae or sister to Rebbachisauridae (see Fig. 6B).

Discussion

Systematics

Dorsal vertebrae PETMG R85

The most notable features on these dorsal vertebrae are the ventral projection of the anterior articular surface, the relative elongation of the centrum when compared to the neural arch, the suggested dorsal projection of the diapophyses by the diapophyseal base, and the nutrient or pneumatic foramina.

The first dorsal centrum furthermore shows mild opisthocoely, and both show a slightly more ventral projection of the anterior articular surface. Opisthocoely in posterior dorsals for instance, is seen in Mamenchisaurus, Omeisaurus and Haplocanthosaurus (Hatcher, 1903; He, Li & Cai, 1988; Ouyang & Ye, 2002) and thus occurs both in non-neosauropod eusauropods and in neosauropods. It should be noted, however, that posterior dorsal opisthocoely has not been found in non-neosauropod eusauropods basal to mamenchisaurids and Omeisaurus, such as Cetiosaurus, Spinophorosaurus, Shunosaurus, Tazoudasaurus, Lapparentosaurus and Patagosaurus (Bonaparte, 1986b; Bonaparte, 1986a; Upchurch & Martin, 2003; Allain & Aquesbi, 2008; Remes et al., 2009), and also not in the isolated Oxford Clay Fm dorsal NHMUK R1986, attributed by Mannion et al. (2013) to a non-neosauropod eusauropod (Figs. 7G–7I). A ventral projection of the anterior articular surface is seen to some extent in Cetiosauriscus (Woodward, 1905) and also in Ferganasaurus (Alifanov & Averianov, 2003).

Figure 7 Comparative schematic drawings of PETMG R85 with posterior dorsals of other sauropods.

The Rutland Cetiosaurus (A), Cetiosaurus oxoniensis (B) and PETMG R 85 (C) in anterior view, and PETMG R85 (D) with Cetiosauriscus (E) and NHMUK R1986 (F) in posterior view. PETMG R85 in lateral view (G) with Cetiosauriscus (H) and NHMUK R1986 (I). PETMG R85 in ventral view (J) with NHMUK R1986 (K). Scalebar is 10 cm, Cetiosauriscus not to scale.

The ratio of centrum dorsoventral length/neural arch length is roughly 4:1, whereas this is roughly 2:1 in Cetiosauriscus (Woodward, 1905), and also in Haplocanthosaurus, and Apatosaurus (Tschopp, Mateus & Benson, 2015), and roughly 1:1 in Cetiosaurus oxoniensis (Upchurch & Martin, 2003). It is likely that the neural arch is incomplete, which gives a disproportionately short length. However, the current measurements prevent these dorsals from being related to Cetiosauriscus.

Pronounced dorsal projection of the diapophyses in dorsal vertebrae is a character shared with Shunosaurus, Cetiosaurus, turiasaurians, Haplocanthosaurus, rebbachisaurids and dicraeosaurids (Hatcher, 1903; Zhang, 1988; Casanovas, Santafé & Sanz, 2001; Upchurch & Martin, 2003; Rauhut et al., 2005) and are thus also present in a wide array of both non-neosauropod and neosauropod dinosaurs (See Figs. 7A–7C).

Small nutrient or pneumatic foramina on the centrum are seen in the dicraeosaurid Suuwassea; however, in this taxon, the foramina express on the anterior caudals (Harris, 2006). Moreover, the lack of any clear pleurocoels on the centra of PETMG R85 might rule out any neosauropod connection. The only dorsal vertebra of Cetiosauriscus shows a small but pronounced pleurocoel (Woodward, 1905).

To summarize, more characters indicative of a non-neosauropod eusauropod origin are present in PETMG R85. Some neosauropod characters exist; however, some of these are also shared with non-neosauropod eusauropods.

Anterior caudal vertebra PETMG R272

The anterior caudal PETMG R272 shows characteristics shared with both non-neosauropod eusauropods, as well as neosauropods.

The slightly more rounded shape of the centrum in lateral view is shared with Apatosaurus. Anterior caudals of Cetiosauriscus are strongly axially compressed, as also seen in non-neosauropod eusauropods such as Cetiosaurus and Patagosaurus (Woodward, 1905; Charig, 1980; Bonaparte, 1986b; Upchurch & Martin, 2003).

The flat anterior articular surface and the mildly concave posterior articular surface of the centrum is a common feature, shared with non-neosauropod eusauropods (e.g., Cetiosaurus, Patagosaurus Bonaparte, 1986b; Upchurch & Martin, 2003). The thick rim cupping the anterior surface is found in early Middle Jurassic non-neosauropod eusauropods (Cetiosaurus) but also in the (non-neosauropod eusauropod/potentially basal neosauropod) Callovian Cetiosauriscus (Woodward, 1905; Charig, 1980; Heathcote & Upchurch, 2003) as well as in the Oxfordian basal titanosauriform Vouivria damparisensis (Mannion, Allain & Moine, 2017). The morphology of the ventrally offset anterior articular surface, together with pronounced chevron facets, is seen in non-neosauropod eusauropods from the Late Jurassic of Portugal (Mocho et al., 2017); however, this type of asymmetry is also seen in Apatosaurus louisae (Harris, 2006).

A ventral keel is found in an Early Jurassic indeterminate sauropod caudal from York, UK (YORYM:2001.9337; Manning, Egerton & Romano, 2015), as well as the Middle Jurassic indeterminate non-neosauropod eusauropod ‘Bothriospondylus’ NHMUK R2599 (Mannion, 2010), and finally, in material ascribed to the non-neosauropod eusauropod Patagosaurus (MACN-CH 232, F Holwerda, pers. obs., 2017). However, this structure is also found in neosauropods, specifically in flagellicaudatans and diplodocids Apatosaurus ajax, Apatosaurus louisae, and the dicraeosaurid Suuwassea (Harris, 2006; Tschopp, Mateus & Benson, 2015). These have a ventral keel which results from a transverse constriction of the ventral side of the centrum, forming a triangular protrusion on the ventral articular surface. This is also seen in non-neosauropod cervicals (such as Cetiosaurus, Patagosaurus, Spinophorosaurus, Amygdalodon, Tazoudasaurus; Bonaparte, 1986b; Rauhut, 2003; Upchurch & Martin, 2003; Allain & Aquesbi, 2008; Remes et al., 2009). The latter keel-like form, which seems to match more with the morphology of PETMG R272, forms when there is a very close association of the two ventrolateral ridges that run along the ventralmost side of the centrum, and is only seen in neosauropods (Harris, 2006; Tschopp, Mateus & Benson, 2015). No keel-like structure is seen in anterior caudals of Cetiosauriscus, nor on the Callovian NHMUK R1984 caudals from the Oxford Clay (Upchurch & Martin, 2003; Noè, Liston & Chapman, 2010); the ventral surface of these anterior caudal vertebrae appearing to be smooth.

The triangular shape of the anterior caudal transverse process (ACTP) complex (Gallina & Otero, 2009) in PETMG R272 is seen to a lesser extent in non-neosauropod eusauropods, such as Tazoudasaurus, Omeisaurus, and Shunosaurus, but also in an unnamed anterior caudal from a possible titanosauriform, but as yet indeterminate eusauropod from the Bajocian of Normandy, France, and in indeterminate non-neosauropod sauropods from the Late Jurassic of Portugal (He, Li & Cai, 1988; Zhang, 1988; Allain & Aquesbi, 2008; Läng, 2008; Mocho et al., 2017). The pronounced shape, however, is more suggestive of ‘wing’-shaped transverse processes, due to the possible existence of a ‘shoulder’ (see Fig. 2). This is used as a caudal character to define diplodocids (Whitlock, 2011; Tschopp, Mateus & Benson, 2015), and is found neither in non-neosauropod eusauropods nor the Bajocian French caudal. However, it is also seen in other neosauropods, such as Camarasaurus and titanosauriforms (Gallina & Otero, 2009). To a lesser extent, a triangular, sheet-like ACTP is seen in Cetiosauriscus (See Fig. 8), as well as the NHMUK R1984 caudals from the Oxford Clay, however, the anterior caudals of Cetiosauriscus do not show a pronounced ‘shoulder’. Moreover, the transverse processes of PETMG R272 are robust, and rounded to triangular in cross-section, whereas those of Cetiosauriscus are gracile, dorsoventrally elongated and axially compressed, providing a more oval cross-section. Though suggestive of a triangular ACTP, the lack of any clear transverse processes on PETMG R272 rule out any firm conclusion on their morphology.

Figure 8 Comparative schematic drawings of PETMG R272 with anterior caudals of other sauropods.

PETMG R 272 in anterior view (A) with Cetiosaurus oxoniensis (B), Cetiosauriscus (C) and an indeterminate non-neosauropod eusauropod from the Middle Jurassic of the UK (YORYM:2001.9337; Manning, Egerton & Romano, 2015), (D). PETMG R272 in posterior view (E) compared to NHMUK R1984 (F) in posterior view (after Noè, Liston & Chapman, 2010). Scalebar 10 cm, Cetiosauriscus and NHMUK R1984 not to scale.

The presence of clearly defined centrodiapophyseal laminae is considered to be a local autapomorphy in the Late Jurassic titanosauriform Vouivria (Mannion, Allain & Moine, 2017). PETMG R272 does show short rugose centrodiapophyseal laminae.

To summarize, more characters indicative of a neosauropod origin of this caudal are present than those indicative of a non-neosauropod (eu)sauropod origin. However, due to the lack of complete transverse processes and neural spine, several morphological characters remain ambiguous.

Middle caudal vertebra LEICT G.418.1956.21.0

The middle caudal LEICT G.418.1956.21.0 also shows characters shared with non-neosauropod eusauropods, as well as neosauropods.

The rhomboid, hexagonal to octagonal shape of the anterior articular surface is not seen in Cetiosauriscus; the middle caudal articular surfaces of the latter are rather round to oval in shape. Hexagonal articular surfaces are a derived condition found in neosauropods, such as Apatosaurus ajax, Suuwassea, but also in Camarasaurus, Demandasaurus and Dicraeosaurus (Upchurch & Martin, 2002; Tschopp, Mateus & Benson, 2015). Mild hexagonal shapes are seen in Cetiosaurus, (Upchurch & Martin, 2003). Octagonal articular surfaces are also a derived feature seen in Dicraeosaurus and the potential neosauropod Cetiosaurus glymptoniensis (Upchurch & Martin, 2003; Harris, 2006).

The anterior placement of the neural spine is another neosauropod character seen in diplodocids and in titanosauriforms (Tschopp, Mateus & Benson, 2015).

The ventrolateral crests seen on the ventral side of this caudal are a neosauropod feature, found in many Late Jurassic neosauropods (Harris, 2006; Mocho et al., 2017). See Fig. 9 for lateral comparisons. The ventral hollow seen in LEICT G.418.1956.21.0 is also found in several neosauropods, such as Tornieria, Diplodocus, Supersaurus, but also Demandasaurus and Isisaurus (Tschopp & Mateus, 2017). However, it is also seen in an unnamed caudal vertebra from the Bajocian-Bathonian of Skye, UK (Liston, 2004). The ventral hollow is also present in Cetiosauriscus (Fig. 9), though not as pronounced as in LEICT G.418.1956.21.0.

Figure 9 Comparative schematic drawings of LEICT G. 418.1956.21.0 with middle caudals of other sauropods.

LEICT G. 418.1956.21.0 in lateral view (A) with the Rutland Cetiosaurus (B), Cetiosauriscus (C) and Cetiosaurus oxoniensis (D). Scalebar 10 cm, Cetiosauriscus not to scale.

The longitudinal ridge is another neosauropod feature, though it is also present in non-neosauropod eusauropods, e.g., Omeisaurus (Ouyang & Ye, 2002). A longitudinal ridge is seen on both Cetiosauriscus and LEICT G.418.1956.21.0 (See Fig. 9), as are the lateral pneumatic foramina on the centra, and the ventrolateral crests.

Nutrient foramina are seen on the Late Jurassic dicraeosaurid Suuwassea (Harris, 2006), but also on Late Jurassic Portuguese non-neosauropod eusauropods; small foramina on the ventral surface of the centrum are also seen in the anterior caudals of non-neosauropod eusauropods from Late Jurassic of Portugal (Mocho et al., 2017).

In summary, more neosauropod characters than non-neosauropod eusauropod characters exist on this caudal centrum; however, as the element is incomplete, the exact placement of this caudal remains uncertain.

Phylogenetic signal and implications for biodiversity

The phylogenetic analysis shows the isolated elements of this study to be unstable OTU’s; in the first analysis based on Carballido et al. (2017), the dorsal elements jump between a position of non-neosauropod to a position nested in Macronaria, with the caudal elements nested a few steps below Camarasaurus. In the second analysis based on Tschopp, Mateus & Benson (2015), the middle caudal element jumps between being sister-taxon to Rebbachisauridae and being nested in Diplodocidae. This, together with the low number of steps needed to break any relationships, shows that the characters on the isolated elements remain ambiguous, as a plesiomorphic array of characters are present. Any implications for sauropod biodiversity in the Peterborough Oxford Clay Formation must therefore be regarded with some caution.

Nevertheless, the possibility exists that in addition to Cetiosauriscus, a neosauropod assemblage (consisting of either diplodocimorph and diplodocid, or rebbachisaurid and diplodocimorph, or macronarian) was present in the Callovian Oxford Clay Formation.

No formal Callovian neosauropod is known thus far, with only derived non-neosauropod eusauropods (e.g., Omeisaurus, Jobaria, Ferganasaurus, Atlasaurus, He, Li & Cai, 1988; Zhang, 1988; Monbaron, Russell & Taquet, 1999; Alifanov & Averianov, 2003; Rauhut & López-Arbarello, 2009) diagnosed. Confirmed neosauropods start to appear in the fossil record in later stages, e.g., from the Oxfordian of France, Vouivria has recently been identified as the earliest titanosauriform (Mannion, Allain & Moine, 2017). The Kimmeridgian-Tithonian fossil record shows neosauropods to be firmly established globally in the fossil record (Mannion et al., 2011, and references therein), with a peak occurrence in diplodocids, macronarians and titanosauriforms from especially the North American Morrison, the Portuguese Lourinhã, and the Tanzanian Tendaguru Formations (including a basal macronarian form from the Kimmeridgian of Germany (Foster, 2003; Remes, 2007; Remes, 2009; Mannion et al., 2012; Mannion et al., 2013; Carballido & Sander, 2014; Mocho, Royo-Torres & Ortega, 2014; Tschopp, Mateus & Benson, 2015)). An early rebbachisaurid has recently been identified from the UK as well; the Early Cretaceous Xenoposeidon (Taylor, 2018), after which rebbachisaurs have been relatively common in Europe and Gondwana (Mannion, 2009; Mannion, Upchurch & Hutt, 2011; Holwerda et al., 2018).

Moreover, early Middle Jurassic neosauropods are possibly present in the Toarcian-Bajocian of Argentina (Rauhut, 2003; Holwerda, Pol & Rauhut, 2015), and Aalenian of China (Xu et al., 2018). Therefore, the presence of Callovian neosauropods present in the UK would not be wholly surprising. Though evidently not as species-rich as the later Kimmeridgian-Tithonian Tendaguru, Morrison or Lourinhã Formation, the Peterborough Oxford Clay material thus far has thus hinted at an equivalent degree of higher level taxonomic diversity to those three classic terrestrial Late Jurassic formations; however, as the material from this study is incomplete, the diagnosis of indeterminate non-neosauropod eusauropod or at best indeterminate neosauropod, is appropriate. Finally, Cetiosauriscus will be revised in the near future (P Upchurch, pers. comm., 2018), therefore further studies on more material may clarify the origin of these remains.

Conclusions

In summary, the associated posterior dorsals show characters shared with both non-neosauropod eusauropods, as well as neosauropods. These elements will therefore be ascribed to an indeterminate non-neosauropod eusauropod. The anterior isolated caudal shares a few morphological features with non-neosauropod eusauropods, and most morphological features with neosauropods. The middle isolated caudal shares a few features with non-neosauropod eusauropods, and more with neosauropods. It is therefore possible that these caudals belong to a neosauropod dinosaur, which are also different to Cetiosauriscus. Phylogenetic analysis tentatively recovers these caudals as neosauropodan. Therefore, these vertebrae give a higher sauropod diversity to the Peterborough Oxford Clay Formation than previously assumed.

Supplemental Information

Supplemental Information 1 TNT files of Carballido et al. (2017) and Tschopp & Mateus (2017)

With added codings of Cetiosauriscus, Cetiosaurus, and PETMG R85, PETMG R272, and LEICT G. 418.1956.21.0.

Click here for additional data file.

The authors would like to thank Glenys Wass and the staff of Vivacity-Peterborough Museum for kindly providing access to the specimen, as well as to the late Arthur Cruickshank of the New Walk Museum, Leicester, for preparing the Leicester material. Furthermore, All McGowan, Tim Palmer, John Cope and Kevin Page are thanked for providing invaluable information on the Oxford Clay invertebrate fossils. Darren Withers helped in identifying the provenance of the Peterborough clay pits. Emanuel Tschopp is thanked for discussion on his dataset. The suggestions and comments by editor Matt Wedel, reviewers Phil Mannion, Darren Naish and one anonymous reviewer greatly improved this paper. We acknowledge the Willi Hennig Society for phylogenetic analysis using TNT.

Institutional abbreviations

PETMG R Vivacity-Peterborough Museum, UK

LEICT G New Walk Museum, Leicester, UK

NHMUK Natural History Museum, London, UK

YORYM York Museums Trust, York, UK

Additional Information and Declarations

Competing Interests

Author Contributions

Data Availability

The authors declare there are no competing interests.

Femke M. Holwerda conceived and designed the experiments, performed the experiments, analyzed the data, prepared figures and/or tables, authored or reviewed drafts of the paper, approved the final draft.

Mark Evans conceived and designed the experiments, analyzed the data, contributed reagents/materials/analysis tools, authored or reviewed drafts of the paper, approved the final draft.

Jeff J. Liston conceived and designed the experiments, contributed reagents/materials/analysis tools, authored or reviewed drafts of the paper, approved the final draft.

The following information was supplied regarding data availability:

Data is available in the Supplementary File and at Figshare: Holwerda et al. (2018) PeerJ Supplemental File. figshare. Fileset. https://doi.org/10.6084/m9.figshare.7302224.v1.

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
