# Peer review of "Additional sauropod dinosaur material from the Callovian Oxford Clay Formation, Peterborough, UK: evidence for higher sauropod diversity"

_PeerJ, doi:10.7717/peerj.6404_

## Round 0.1 · original submission · Major Revisions

Congratulations - all three reviewers found value in this work, and all three are positive about its potential for publication following revision. That said, the reviewers make numerous constructive suggestions for improvement.

Reviewer 1 would prefer to see a more inclusive matrix used for the phylogenetic analysis, whereas Reviewer 3 finds it unnecessary given that the current matrix is focused on diplodocoids. I recommend you take Reviewer 1's suggestion and make the analysis more relevant by broadening it to include more non-diplodocoids, especially basal sauropods and basal eusauropods. I also agree with Reviewer 3's suggestion that more and better-illustrated comparisons to Middle Jurassic material from the British Isles would strengthen the paper.

Those are only the most noteworthy suggestions by the reviewers - I've read them carefully and find them all to be pertinent. Please be diligent in addressing all of the issues raised. I will look forward to seeing an improved version of this work soon.

·

Basic reporting

The paper is generally well-written and structured. There are a number of unsubstantiated statements in the Introduction which require adding in some references, and some identifications of previously described remains are outdated (i.e. there is more recent literature re-interpreting the affinities of these elements).

Experimental design

No comment

Validity of the findings

Please see General comments for the author

Additional comments

This manuscript describes two previously unpublished sauropod caudal vertebra from the late Middle Jurassic of the UK. Our knowledge of sauropods from that time interval is generally poor, and so although these specimens are incomplete, this is still a welcome contribution. Currently we have no unambiguous evidence for the derived sauropod clade Neosauropoda in sediments older than the Late Jurassic, and yet this MS presents evidence that these two caudal vertebrae belong to the neosauropod clade Diplodocoidea. I think that the Description and Figures are good and detailed, and the paper is generally well-written, but I have a number of issues with the content. The main ones are summarised below, and I have also provided an annotated PDF of the MS.

(1) I think the authors should be cautious about interpreting these remains as a neosauropod, given that we have no unambiguous record of this clade pre-Late Jurassic (see comment #2), and that the two vertebrae are highly incomplete. Although some features do indeed seem to suggest that these remains could represent a neosauropod (e.g. a ventral keel and shallow lateral fossa on the anterior caudal; ventrolateral ridges on middle caudal centra), I would still be a little wary of this interpretation. An anterior caudal centrum of the Middle Jurassic non-neosauropod eusauropod ‘Bothriospondylus madgascariensis’ also has a subtle midline ventral keel and lateral fossa, for example. The Chinese Middle Jurassic mamenchisaurid Chuanjiesaurus also has ventrolateral ridges in some of its middle caudal centra.

One feature used to support diplodocoid affinities is: “the high projection on the neural arch of the diapophyseal laminae suggest the existence of a ‘shoulder’, which would make the transverse processes wing-shaped”. However, I don't think that you can make this inference based on what's preserved: for it to be wing-shaped, the dorsal margin of the rib should be deflected such that it has a dorsolaterally oriented dorsal margin.

In addition, the data matrix used is primarily focused on evaluating the phylogenetic interrelationships of diplodocid sauropods, and so this is not really the most appropriate matrix to use to try and determine the placement of new specimens. It would be much better to use a matrix that includes a wide array of sauropod taxa (and thus characters) from across the tree (e.g. the most recent Carballido matrix [Patagotitan]). If analysis of that matrix indicates diplodocoid affinities, then it would be appropriate to incorporate it into a diplodocoid-focussed matrix. Finally, if you are to include it in a diplodocoid-focussed matrix, the more recent Tschopp and Mateus (2017) matrix (Galeamopus paper) would be better as it includes a wider array of diplodocoid taxa, as well as updated scores.

I should clarify that I have no problem with neosauropods being present in the Middle Jurassic: these newly described remains might well belong to that clade. Given that we have derived members of Neosauropoda distributed fairly globally in the Late Jurassic, it’s almost certain that the clade radiated prior to the Late Jurassic. But we should be careful when extending this range back in time, especially when based on very incomplete material.

(2) Throughout the MS, previously described Middle Jurassic sauropods are stated as representing neosauropods (including specimens from the same stratigraphic unit as the newly described vertebrae), but nearly every single instance has been refuted in the literature over the last decade. As such, the authors should be careful that they are reflecting the most recent taxonomic opinions on these often fragmentary specimens. The updated IDs are commented upon in the annotated PDF.

(3) In several places it’s stated that caudal vertebrae have been neglected in phylogenetic studies, e.g. in the Introduction: “Moreover, caudal vertebrae have rarely been given appropriate attention, as only recently have caudal characters begun to be recognized as taxonomically diagnostic (e.g. Mocho et al., 2017; Holwerda & Liston, 2017)” I don't agree with this at all. I think if you look at character lists in recent papers by Carballido, D'Emic, Tschopp, Whitlock and myself (as well as older papers by Wilson, Upchurch, and others), you'll see that there's actually quite a lot of caudal vertebral characters out there, and that such emphasis on this part of the vertebral column didn't begin in 2017. The last sentence of the Discussion states: “This shows the importance of paying close attention to caudal characters, as phylogenetic information might otherwise be missed…”. However, no new caudal characters are devised in this work, and the very fact that you're able to resolve the position of these vertebrae in an existing data matrix, without adding further characters, runs contra to this argument that this work somehow captures information previously missed.

(4) One of the conclusions of this MS is that these and previously described specimens from the Oxford Clay indicate "an equivalent richness of sauropod groups from this Middle Jurassic marine formation, to the classic Late Jurassic terrestrial Morrison, Lourinhã and Tendaguru formations”. This is a bit of a leap! The Oxford Clay has Cetiosauriscus and then a bunch of additional indeterminate remains, at least some of which are different to Cetiosauriscus. But there isn’t evidence for a brachiosaurid (anteriormost caudals) and camarasaurid (a tooth) in the formation – those are outdated identifications (see point 2) – and that distal caudal sequence cannot be confidently attributed to a diplodocid. Furthermore, although the two vertebrae described in this contribution come out in different places in the tree, this doesn’t necessarily mean that they represent different taxa: they’re not overlapping, which makes it less likely that they would cluster together, and certain derived features might only be present in the middle caudal. If you split up any fragmentary taxon and scored two it's vertebrae as separate OTUs they probably wouldn't cluster together. So, I don’t think you have good evidence for lots of taxa in the Oxford Clay – taking a conservative measure, maybe three at most (Cetiosauriscus and two indet. taxa, one of which might be a diplodocoid).

(5) Finally, there are several statements in the Introduction about the timings of sauropod diversifications and the group’s fossil record, but none of these are supported with references.

Best wishes,

Phil Mannion

·

Basic reporting

This is a nice, tidy analysis of some interesting isolated specimens: the study is succinct and the analysis and conclusions are well explained and done well.

Experimental design

Everything seems fine and satisfactory.

Validity of the findings

Everything seems clear, well explained etc. See general comments for more.

Additional comments

After analysing the specimens concerned, the authors conclude that several different taxa – belonging to markedly different positions within Sauropoda – are represented. This conclusion may well be valid. But – given that we are talking about isolated caudal vertebrae alone – I would expect there to be at least some brief caveat crediting the possibility that the specimens might not carry sufficient phylogenetic value to justify these conclusions. In other words, there should – in my opinion – be a brief discussion where the authors either add caveats, or credit the possibility that the specimens might possibly be misleading as goes their phylogenetic position, and that one or more of them might actually belong to the same taxon.

The manuscript needs some minor tweaking as goes grammar and sentence structure. There are, for example, a few places where a semi-colon is needed instead of a comma (e.g., line 79), and quite a few sentence where re-wording and re-structuring is advised (e.g., lines 82-86, lines 89-91).

Line 34: I am slightly uneasy these days as goes referring to a taxon as ‘derived’; it has now become shorthand for ‘advanced’ and is not what the authors mean anyway. What they mean, specifically, is that the taxon ‘belongs in a deeply nested position within the clade in question’ (this being Eusauropoda, in this case). See further comments later in this review.

Line 63: taxonomic pedantry… given that pliosaurs are part of Plesiosauria, it can be argued that talking of ‘plesiosaurs and pliosaurs’ is unsatisfactory, and that better wording would be something like ‘pliosaurids, cryptoclidids and other plesiosaurians’. Also, ‘marine crocodylomorphs’ is better than ‘marine crocodiles’ (the animals concerned are not ‘crocodiles’ at all).

Line 87 (and also line 439): the authors who write about Turiasaurus and kin have named the respective clade Turiasauria, not Turiasauridae, meaning that the vernacular term should be either ‘turiasaur’ or ‘turiasaurian’.

Linex 98-100: I don’t think it’s at all true that caudal vertebrae have only recently “been begun to be recognised as … diagnosable”. Look, for example, at the several studies on titanosaurian caudals [Le Loeuff on Iuticosaurus, the discussion provided by Wilson and Upchurch 2003 and so on] and their claimed value in distinguishing taxa. Or… are you specifically referring to material from the Middle Jurassic here?

Line 151: double space present.

Line 164: ‘shallowly concave’ or similar might be better than ‘mildly’ concave.

Line 215: do you need to say that it is “an isolated element” given that the paper is about isolated caudal vertebrae?

Line 286: as a vernacular term, mamenchisaurids should be written with lower case first letter. Rather than describing the specimen as ‘more derived than mamenchisaurids etc..’ (since ‘more derived’ has connotations of ‘more advanced than’), it might be better to describe the specimen’s position with regard to which clade it belongs to. But, in any case, does this description match what you show in the tree? In Fig 4, Cetiosauriscus is not “more derived than mamenchisaurids”: it is part of the same clade, and in fact is a mamenchisaurid according to some phylogenetic definitions of that clade. Or have I missed something? Given that Cetiosauriscus is recovered here as a possible mamenchisaurid, it might be appropriate to cite other studies that found this too: Rauhut et al. (2005) did so, and Naish and Martill (2007) discussed this possibility as a result of Rauhut et al.’s (2005) conclusions. Naish and Martill (2007) should probably be cited somewhere in the present study, given the reviews provided there of Cetiosauriscus and other Oxford Clay sauropods.

Lines 288-290: you describe the position of R272 as being “more derived than Haplocanthosaurus, but basal to Zapalasaurus and all derived neosauropods”. Firstly, I would recommend against saying that any phylogenetic position makes a specimen or taxon “more derived” than another – you mean, instead, that the specimen is more nested within the given clade (Diplodocoidea) than is Haplocanthosaurus. In this case, I would say something like “R272 is recovered as being closer to diplodocids than is Haplocanthosaurus, but outside the clade that includes rebbachisaurids and remaining diplodocoids”. There is a trend in tree descriptions to prefer wording of this sort, and to refer more to tree shape, sister-group relationships and clade membership. Secondly, you’ve recovered R272 as a diplodocoid here (since Diplodocoidea is defined as the branch-based clade that includes all neosauropods closer to Diplodocus than to Saltasaurus… you didn’t include Saltasaurus, but I think we have to assume that Saltasaurus and other titanosaurs would be on the same lineage as Brachiosauridae. You might want to comment on this). In describing R272’s position, you should therefore be describing its position relative to other diplocoids, not other neosauropods.

Lines 295-297: similar comments apply here. Rather than describing the position of G.418.1956.21.0 as “more derived than” etc and “basal to…” etc, I would recommend something like “ we recover G.418.1956.21.0 as part of a clade that includes Barosaurus and Diplodocus but excludes Galeamopus, and as the sister-taxon to a Barosaurus + Kaatedocus clade”. Note that you have not found to be “more basal” than “Barosaurus and all other diplodocids”, as per your description of the tree.

Line 332: I think it’s more typical to use ‘flagellicaudatans’.

Line 413: ‘recovers’ is a better term than ‘retrieves’.

References

Naish, D. & Martill, D. M. 2007. Dinosaurs of Great Britain and the role of the Geological Society of London in their discovery: basal Dinosauria and Saurischia. Journal of the Geological Society, London 164, 493-510.

Rauhut, O. W. M., Remes, K., Fechner, R., Cladera, G. & Puerta, P. 2005. Discovery of a short-necked sauropod dinosaur from the Late Jurassic period of Patagonia. Nature 435, 670-672.

Wilson, J. A. & Upchurch, P. 2003. A revision of Titanosaurus Lydekker (Dinosauria – Sauropoda), the first dinosaur genus with a ‘Gondwanan’ distribution. Journal of Systematic Palaeontology 1, 125-160.

Reviewer 3 ·

Basic reporting

Dear Editor and authors.

It was a pleasure to review the present manuscript entitled “Additional sauropod material from the Peterborough Oxford Clay: evidence for higher sauropod diversity”, and in the attached pdf you can access part of my comments and suggestions. The English seems ok for me.

The sauropods of the Middle Jurassic are particularly unknown and any information can be valuable in order to understand these fauna. Furthermore, the dinosaur material from UK have been a classic disaster, taxonomically, and it is great to have people working on that. However, I believe this work fails on the approach. The material is too fragmentary, and the characters used to support diplodocoid affinities are not well-supported, and in some cases, not properly interpreted (in my opinion). A more accurate comparative study with the Middle Jurassic specimens of England is needed, and a clear comparison with key specimens, such Cetiosaurus oxoniensis, Rutland sauropod and Cetiosauriscus are necessary to really improve our knowledge on the sauropods from the Middle Jurassic of England, and consequently, the present manuscript (figuration could be a great improvement, including more specimens and being a more broader view about the caudal anatomy shown by sauropods of the Middle Jurassic of England, e.g., Cetiosaurus leedsi, rutland specimen, the posterior caudal of the NHMUK). I also think that is important to have a more conservative position counting the possible number of taxa. The comparison with the Morrison, Tendaguru and Lourinhã needs to be developed and explained in a better way. "Cetiosaurus leedsi" is different from Cetiosauricus and Cetiosaurus oxoniensis?

The phylogenetic analysis is for me unnecessary, especially when you are using a data set specifically focused on diplodocids. Improve your comparative study. Did you have good characters to include in future phylogenetic analyses, e.g. the deep crpf are curious? the heart-shaped morphology of the anterior articulation can be traced phylogenetically? In conclusion, do you have some good tail specimens in the Middle Jurassic, used them.

And please, reconsider the orientation of the anterior caudal vertebra described herein.
The incorporation of more specimens will be important in order to justify this paper in a PeerJ.

I will suggest this paper for publication with major revisions. I will be glad to review a second version. I really see potential in this work

Sincerely

Experimental design

See above

Validity of the findings

See above

Annotated reviews are not available for download in order to protect the identity of reviewers who chose to remain anonymous.

---

## Round 0.2 · Minor Revisions

I'm happy to report that both reviewers found the manuscript improved. There are still some areas that require attention. In particular, I share Reviewer 1's skepticism about the possible parapophyses down at the centrum/arch junction in the dorsals--mostly, I see no very convincing evidence that those are in fact parapophyses. Reviewer 1's points in the discussion regarding evolution and biogeography are also apposite and require attention. Reviewer 2 made many helpful suggestions toward a cleaner and tighter manuscript.

One minor thing: the structure identified as a fossa on the ventrolateral aspect of the centrum in Figure 5F looks like it might possibly be damage from crushing--can you confirm one way or another?

Please carefully consider all of the reviewers' comments and revise the manuscript accordingly, and I will look forward to seeing an improved version soon.

·

Basic reporting

Overall I don't have much to comment here, but I think that the Description might benefit from some subheadings to separate each set of elements (i.e. Dorsal vertebrae; Anterior caudal vertebra; Middle caudal vertebra).

Experimental design

No comment

Validity of the findings

Please see General comments for the author

Additional comments

The manuscript is greatly improved from the original submission, but I still have some problems with aspects of the revised MS. These are provided as an annotated version of the MS (including typos and grammatical errors), and below I have summarised some key issues:

(1) A couple of identifications in the Description are problematic. One of the dorsal vertebrae is described as having parapophyses situated on the very base of the neural arch (partly on the centrum), and the other is annotated with the parapophysis in the same place. However, they are interpreted as posterior dorsal vertebrae. The parapophyses on middle-posterior dorsal vertebrae of all sauropods are situated high on the neural arch. If correctly identified in the newly described dorsal vertebrae, they must be anterior dorsal vertebrae, approximately Dv3. This would also affect many character scorings, as most dorsal characters pertain to middle-posterior dorsals. Less of a problem, but the tentative identification of prespinal and postspinal laminae in the anterior caudal vertebrae is also very unlikely - if correct, this would mean that these laminae extend all the way down to the neural canal opening, and there would essentially be no neural arch region. As the authors note, there is another possibility, i.e. that they represent a TPRL/TPOL, hyposphene, and I think that they should go with this as a much more plausible identification.

(2) The MS seems a bit confused by exactly what the conclusions are regarding the identification of these vertebrae. Sometimes they're definitely neosauropods, whereas at other times they might be, and at other times it's very ambiguous. I would recommend plumming for the middle option, and standardising this throughout the MS.

(3) The final section of the Discussion ('Phylogenetic signal and implications for biodiversity') contains a number of claims for other Middle Jurassic neosauropods (lines 514-515). I wrote comments about the updated ID of two of these last time, and pointed out that the third reference didn't state anything about neosauropods, and yet these points have been completely ignored (they also weren't refuted in the response document). I've repeated my comments in the annotated MS. Lines 517-521 briefly comment on other taxa, but it's unclear what the focus of these sentences is - it seems like a random assortment of taxa from various geographic regions.

Best wishes,

Phil

·

Basic reporting

The manuscript is well presented, is well written, and fairly cites existing literature and the work of others.

Experimental design

The project's aims are clearly stated and the relevant methods used are well defined.

Validity of the findings

The conclusions are well stated and obvious and the article represents a valid contribution to knowledge.

Additional comments

Lines 97-100: the sentence structure here is complex and, I suggest, needs revision. I would say: ‘In addition to Cetiosauriscus, four anterior vertebrae are known from the Oxford Clay Formation. These were previously ascribed to a brachiosaurid, but were more recently reidentified as an indeterminate non-neosauropod’.
Line 101, the sentence should be started with something different from ‘Also’.
Line 192: shouldn’t the ammonite be a common ammonite _of_ the Oxford Clay Formation?
Line 203: I feel that better wording would be ‘… is known. However, the King’s Dyke pit…’.
Line 209: remove ‘back to’.
Line 322: did you really mean ‘bone matrix’ here?
Lines 522-525: I am slightly uneasy as goes the use of ‘nested with’ and ‘nested in’. When describing phylogenetic results, I personally would only use ‘nested’ when referring to an OTU that is found to be deeply embedded within a given clade; what is being described here is a simple grouping with a given OTU or membership of it. I would therefore say.. ‘recovers four trees where PETMG R272 groups with Cetiosauriscus in Diplodocimorpha…’, and ‘… being within Diplodocinae or sister to Rebbachisauridae’ (line 524-525). Also, given that PETMG R85 and LEICT G.418.1956.21.0 are not named, it might be better to refer to (in line 523 and 55) them as ‘sister to Diplodocidae/Rebbachisauridae’ (respectively), not ‘sister-taxon’.
Lines 645-646: the Mannion et al. citation has been accidentally italicised.
Line 655: remove comma after ‘present’.
Line 732: it may be appropriate to cite Taylor 2017 on Xenoposeidon, since it now appears to be an additional Early Cretaceous rebbachisaurid, and possibly an especially early one…
Taylor, M. P. 2017. Xenoposeidon is the earliest known rebbachisaurid sauropod dinosaur. PeerJ PrePrints 5: e3415.
In Fig 6, some clade names are written in technical style (eg, Diplodocoidea), and others in vernacular style (eg, Rebbachisaurids). I assume that all of these terms should be written as technical names.

Darren Naish
[email protected]

---

## Round 0.3 · accepted · Accept

Thank you for your diligence in addressing the reviewers' concerns. I am happy to accept your manuscript for publication in PeerJ.

There are a few minor things you should consider while in the production phase. First, a couple of minor technical points that I had not spotted before (all line numbers refer to the reviewing PDF, not the Word doc):

- lines 399-401: dorsally projecting diapophyses are also present in rebbeachisaurids
- lines 537-539: Xenoposeidon may not be the oldest rebbachisaurid if Maraapunisaurus (formerly Amphicoelias fragillimus) is a rebbachisaur as proposed by Carpenter (2018); it would be safer to refer to Xenoposeidon as "an early rebbachisaurid" rather than "the earliest rebbachisaurid"

There are several minor grammatical and punctuation issues that require attention:

- line 72: would more typically be given as Neosauropoda indet., with the formal clade name and with a period after indet.
- lines 72, 345, 348: unnecessary comma after et al. or before parentheses; there may be more cases but these are the ones that caught my eye
- lines 142-143: "in addition there would also be the worked out pits that would be accessible for collectors to search the pit faces and spoil heaps of." I believe this would read better as a separate sentence, "Collectors would also have had access to the pit faces and spoil heaps of worked-out pits"
- line 384: 'occur' should be 'occurs'
- line 389: 'wel' should be 'well'
- line 391: 'extend' should be 'extent'
- line 397: 'offset' is not the word you want here; perhaps 'prevent' or 'obviate'?
- line 404: 'is seen' should be 'are seen'
- lines 543-544: I believe you intend "not wholly surprising" rather than "not wholly unsurprising"; the double negative in the latter implies suprise!

I recommend the old trick of reading the entire manuscript aloud to discover any other minor issues that you may wish to address before submitting your final version for publication.

The decision of whether or not to publish the peer reviews alongside the paper is entirely yours, and will not affect how your paper is handled going forward. However, I encourage you to do so. Making the reviews public allows the reviewers to receive credit for their efforts, and also contributes to the emerging culture of fairness and transparency in editing and peer review.

#